# Triadic relationships between pasture exposure, gastrointestinal parasites, and hindgut microbiomes in grazing lambs

Jack Jefferson[1]☯, Claire Reigate[2]☯, Alessandra Giacomini🄾[1], M. Jordana Rivero[2], Matthew Hitchings[3], Tamsyn Uren Webster[1‡], Konstans Wells🄾[1‡*]

1 Department of Biosciences, Swansea University, Swansea, United Kingdom, 2 Net Zero and Resilient Farming, Rothamsted Research, North Wyke, Okehampton, United Kingdom, 3 Medical School, Swansea University, Swansea, United Kingdom

☯ These authors contributed equally to this work.
‡ TUW and KW also contributed equally to this work.
* k.l.wells@wansea.ac.uk

## Abstract

Livestock grazing in confined pastures often means grazing on a less diverse diet than under more natural conditions and increased exposure to gastrointestinal parasites prevailing in these pastures. However, how sward composition influences gut microbiome (GM) diversity and its relationship with parasite burden remains poorly understood. In this study, we analysed the faecal GM of weaned lambs grazing on two distinct sward types (perennial ryegrass and a mixed-species sward) over three consecutive months using 16S rRNA sequencing, in order to assess how microbial diversity and composition are related to environmental conditions and the gastrointestinal nematode (GIN) burden in naturally infected lambs. Sward type and sampling time explained some of the variation in GM alpha diversity and community composition (beta diversity), whereas individual lamb identity accounted for considerably more variation in microbial assemblages. Shifts in the relative abundance of bacterial genera such as *Saccharofermentans, Anaerosporobacter, Butyrivibrio* in relation to sward type and sampling time suggest mostly adaptive fluctuations in response to diet and pasture condition. Abundance shifts of *Negativibacillus,* and *Candidatus Saccharimonas* were also associated with GIN burden, which, in turn, was higher in lambs grazing on mixed swards compared to ryegrass. Our findings add to the growing understanding of how sheep microbiomes vary with pasture management and changes in parasite burden. We highlight that individual identity may shape gut microbiota, and that potential triadic interactions among gastrointestinal parasites, sward exposure, and the gut microbiome underscore the importance of considering host, parasite, and environmental factors collectively when evaluating microbiome dynamics in grazing livestock.

**Data availability statement:** Sequence reads have been deposited in the European Nucleotide Archive under accession number PRJEB93828 (ERP176703) (https://www.ebi.ac.uk/ena/browser/view/ PRJEB93828). The metadata and the computer code used for data compilation/manipulation and analysis is available from Github at https://github.com/konswells1/Lamb-microbiome-study/.

**Funding:** We thank the Ecological Continuity Trust for a grant that supported the laboratory work and the BSAS Steve Bishop Net Zero Award for supporting field research and parasitological work. We thank the Rothamsted Farm Staff for support during field work with support from Techion UK Ltd. The grazing experiment is part of the Institute Strategic Programme "AgZero+: Towards sustainable, climate-neutral farming" (NE/W005050/1), an initiative jointly supported by NERC and BBSRC. The funders had no role in study design, data collection and analysis, decision to publish, or preparation of the manuscript.

**Competing interests:** The authors have declared that no competing interests exist.

## Introduction

Livestock grazing in an industrialised world requires efficient and sustainable production in order to meet the demand of global markets and minimise environmental impacts. In livestock production systems, grazing in confined pastures often means foraging on managed pasturelands that are often less diverse in plant species than natural grasslands [1,2]. The spatiotemporal clustering of livestock on confined pastures can increase exposure to gastrointestinal parasites, which pose a major constraint to efficient livestock production by impairing animal health, welfare, and performance, thereby undermining the sustainability of production systems [3,4]. To mitigate these challenges, previous studies have suggested that incorporating multifunctional, bioactive swards can enhance livestock performance [5] and reduce dependence on anthelmintic treatments, particularly in lambs [6,7]. A deeper understanding of the interplay between pasture plant species composition and host–parasite–microbiome interactions is therefore essential for advancing sustainable and resilient livestock production systems and to helping balance agricultural intensity with biodiversity conservation.

Host-associated microbiomes may mediate interactions between diet, animal health, and parasitic infections. In recent years, there has been growing interest in the role of the gastrointestinal microbiome – both in the rumen, the primary site of fermentation in ruminant livestock, and in faecal communities, which serve as a non-invasive proxy for the lower gastrointestinal tract – in influencing animal health, nutrient absorption, greenhouse gas emissions, and the modulation of parasite burdens [8–10]. Intestinal microbiomes are integral to digestion and nutrient uptake [11,12], as well as modulating the host immune response and resistance of animals against parasite infections [13,14]. Generally, for symbiotic microbial communities associated with a host organism, the ecological niche is jointly structured by host intestinal features, social interactions, and external mediators such as diet, environmental stressors, pathogen and parasite exposure, or the application of drugs such as antimicrobials [15–18]. Moreover, exposure to microbes from the environment is critical for microbial community assembly and stability [17]. While most attention in ruminants has focused on the rumen, the hindgut microbiome is also of significant interest given that microbial activity allows the digestion of diet components that reach the hindgut [19]. Environmental conditions linked to diet and rearing conditions can, for example, shape the hindgut microbial assemblage in the early life of ruminants [20,21].

Several studies have reported strong associations between the composition of the gastrointestinal microbiota in grazing herbivores, including both ruminants and hindgut fermenters, and the type of grassland grazed [22–24]. Across a range of vertebrate host species, growing evidence indicates that gastrointestinal helminth infections can influence both the diversity and composition of the gut microbiome [25]. However, the outcomes of helminth–microbiome interactions are variable. Some studies report increased microbial diversity, others observe reductions, and many highlight shifts in the relative abundance of specific bacterial taxa associated with helminth infection, such as reduced *Bacilli* abundance in association with *Hymenolepis*

diminuta infections in rats [26], or increased *Lactobacilli* abundance following *Trichuris suis* infection [27]. In sheep, experimental infection with *Haemonchus contortus*, a highly pathogenic gastrointestinal nematode (GIN), has been associated with changes in gut bacterial composition in relation to GIN burden [14]. These findings suggest that parasite–microbiome interactions are likely dynamic and bidirectional, with parasites influencing microbial communities, and microbial states potentially affecting parasite establishment, survival, or pathogenicity. However, despite recent advances, empirical evidence on how different pasture types influence microbiome-parasite dynamics in sheep and other livestock under natural grazing conditions remains underexplored.

In this study, we analysed the faecal gut microbiome (GM) of weaned lambs grazing two sward types (a mixed sward and a perennial ryegrass-based permanent pasture) across three consecutive months, using temperate grasslands in the UK as a model system. Our aim was to investigate how microbial diversity and composition are influenced by spatiotemporal environmental variation and GIN burden in naturally infected lambs.

## Materials and methods

### Study site and sample collection

The study took place on the Rowden plots at Rothamsted Research in Okehampton (50°46'10" N, 3°54'05" W), Devon, UK. The Rowden plots were established in 1982 as 12 long-term experimental grassland plots of one hectare each, used for grazing and allowing to study the effect of different factors such as drainage, grazing management and different plant biodiversity on grazing systems performance [28]. Each plot was stocked with 20 individually marked Charolais×Suffolk mule crossbred lambs (*Ovis aries*) born in April 2023 that were weaned just prior to the start of sampling. Prior to weaning, lambs had been grazing the same plots with their dams from 2–3 days of age. For this study, we focused on lambs from four experimental plots that were characterised by two distinct sward types, with each type established on two separate plots: 1) 'ryegrass': perennial ryegrass-based permanent pasture, *Lolium perenne* L. (representing the most commonly grown production grass in the UK), or 2) 'mixed sward': ryegrass enriched with other plant species sown in summer 2022 as a seed mix of festulolium (*Festulolium* cv. *Lofa*), timothy (*Phleum pratense*), red clover (*Trifolium pratense*), white clover (*Trifolium repens*), chicory (*Cichorium intybus*), ribwort plantain (*Plantago lanceolata*).

We analysed 60 freshly voided faecal samples collected non-invasively from 21 weaned, individually marked lambs grazing on four experimental plots. Samples were collected on three dates between July and September 2023 (20th July, 20th August, 13th September), with 21 samples each in July and August, and 18 in September. While the same marked lambs were repeatedly sampled over time, the final dataset comprised samples randomly selected from a larger pool of individuals, and therefore did not include exactly the same individuals across all sampling occasions. The whole faecal pat was collected and put into a sterilised plastic sample pot and transferred back to the lab. In the lab, three sub-samples (~50 mg) were collected from the interior of each faecal pat and transferred to 2-ml microcentrifuge tubes pre-filled with RNAlater. These were kept at room temperature for 24 hours, then stored at −20°C until further analysis.

The lambs were weighed and health checked fortnightly (as part of additional research projects), the weights at weaning ranged from 18.5 to 38 kg (mean of 30 kg ± 1.1 one standard error, SE), the daily live weight gains of the lambs during the sampling period ranged from 0.142 to 0.331 kg per day (with no apparent differences in weight gain among sward types). Due to a *Haemonchus* outbreak at the study farm, all lambs were treated with moxidectin (oral dose of 1% Cydectin®) one day before (lambs on two ryegrass and one mixed sward plot) or within a week after (lambs on one mixed sward plot) the first fecal sampling event in July. Six lambs were also treated with albendazole (Ovidrench®) due to high faecal egg counts (FEC) and poor performance. We anticipate that these treatments likely influenced parasite counts and may have unknown effects on the microbiome, although previous work suggest that gut microbiota of treated sheep may return to control level within <1 months after anthelminthic treatment [29]. However, by focusing on relative changes in microbiome composition and parasite–microbiome relationships, we aim to provide meaningful insights, while recognising the

limitations of the study. This non-invasive study was approved by the Ethical Review Committee of Rothamsted Research (North Wyke, United Kingdom) under Project License number P592D2677 and was conducted in accordance with the Animal Scientific Procedures Act (1986) Amendment Regulations (2012).

## Gastrointestinal nematode faecal egg count

The faecal samples were tested for GIN FEC using the FECPAK[S5] method (Techion Group Ltd., NZ). For this, 5 g of sample was added to 25 ml of water and this was homogenised to remove any lumps. Then, 12 ml of slurry were transferred into a 15-ml test tube and centrifuged for 2 minutes at 1,500 rpm. The supernatant was discarded, and the pellet was resuspended in saturated saline (Specific Gravity 1.3). This was then transferred into a pliable centrifuge tube and centrifuged again for 2 minutes at 1,500 rpm. Then, using metal forceps, 1 cm from the top of the centrifuge tube, it was squeezed and the solution above the forceps was transferred into a clean 15-ml test tube. This was filled to 9 ml using saturate saline. The solution was homogenised and 450 µl were transferred to each well in a FECPAK cassette and left to stand on the bench for 6 mins, before being loaded into the FECPAK Micro-i digital imaging device. Observed strongyle eggs on the images were counted using the markup function of the FECPAK Lab software. Using this method, we obtained FEC as relative abundance counts for *Nematodirus battus* (*Strongyloidea*, *Molineidae*). All other nematode eggs were pooled into a single strongyle-type FEC category (Strongyloides and related genera), because most could not be reliably identified to the species level. *Coccidia* oocysts and tapeworm eggs were also noted if present (but not considered in this study).

## Microbiome extraction, sequencing and bioinformatic processing

Total genomic DNA was extracted from faecal samples using the prepGEM Bacteria kit (MicroGEM, Charlottesville, VA, USA), according to the manufacturer's instructions, with an additional preliminary wash step using 10% bleach concentration to minimise potential surface contamination. After the decontamination step, we pooled and homogenised the material from the three sub-samples for each individual sample using 0.2-mm ceramic beads before extracting the DNA.

We performed a two-step polymerase chain reaction (PCR) targeting the V4 region of 16S rRNA gene using the updated sequences of the 515F and 806R primers [30,31], encompassing Illumina overhangs for index barcoding. The first PCR reaction of 20 µl consisted of 2 µL of DNA, 10 µL of Platinum™ II Hot-Start PCR Master Mix (2X) (Thermo Fisher Scientific, Waltham, MA, USA), 0.4 µL of forward and 0.4 µL of reverse primers and 7.2 µL of ultra-pure water. Reaction conditions consisted of an initial denaturation at 95 °C for 3 min, followed by 28 cycles of 30 sec at 95 °C, 30 sec at 55 °C, and 30 sec at 72 °C, and finally 72 °C for 5 minutes.

During the second PCR, sample-indexing was performed using the Nextera ® XT Index Kit (Illumina, Inc., San Diego, CA, USA), in a 25 µL reaction consisting of 2.5 µL of amplified DNA from the previous PCR, 12.5 µL of Platinum™ II Hot-Start PCR Master Mix (Thermo Fisher Scientific, Waltham, MA, USA), 1.25 µL of each index, and 10 µL of ultra-pure water. Reaction conditions were as above, but with 8 cycles instead of 28. PCR products were pooled based on agarose gel band relative intensity, cleaned using AMPure XP beads kit (NEBNext® sample purification beads, USA) according to manufacturer's instructions, and quantified via qPCR using the NEBNext® Library Quant Kit for Illumina® (Ipswich, MA, USA). Final libraries were normalised to 4 nM and sequenced on an Illumina MiSeq platform (paired 300 bp reads). Two negative controls (PCR library blanks) were prepared and sequenced alongside the faecal DNA samples. The negative controls each yielded < 300 reads and were therefore excluded from subsequent analysis.

Sequence reads were processed using QIIME2 2024.5.0 [32]. Raw sequences were truncated at 230 bp (forward) and 160 bp (reverse) and trimmed (leading 19 bp) to remove primer sequences. Amplicon sequence variants (ASVs) were obtained following denoising and chimeras removal using the DADA2 plugin (Callahan et al., 2016). The ASV taxonomy was classified by training a naïve-Bayes classifier on the SILVA reference database in release version v.138 [33] for 16S rRNA gene sequences, and ASVs assigned to chloroplasts or mitochondria were removed. Taxa names were checked

for matches with the National Center for Biotechnology Information (NCBI) database, using the *taxize* 0.9.1 R package [34] and the number of identifiable taxa at the phylum, family, and genus level based on these matches reported. Raw sequence reads have been deposited in the European Nucleotide Archive under accession number PRJEB93828 (ERP176703) (https://www.ebi.ac.uk/ena/browser/view/ PRJEB93828). The final ASV count data consisted of 3,637,046 reads from the 60 gut microbiome (GM) samples.

## Data analysis

Statistical analyses were carried out in R software version 4.3.2 [35], using the *ggplot2* package for graphical display. For alpha diversity analysis, we iteratively rarefied samples (n = 100 times) to a depth of 11,531 reads according to control for uneven library size in our data. For each rarefied subset, we then computed Chao1 species richness estimates (as measures of relative species richness for the standardised library size) and Shannon diversity estimates (accounting for the evenness in ASV abundances) using the *vegan* package [36]. We used generalized linear mixed models (GLMMs, Gaussian distribution; *glmmTMB()* function from *glmmTMB* package [37]) to explore possible differences in alpha diversity in relation to sward type, time of sampling (both implemented as categorical covariates) and FEC of *Nematodirus battus* and strongyles (used in model as z-scores of the log-transformed FEC, following in R the notation "scale(log(x+0.001)"). The plot IDs was included in the models as random effects to account for repeated measures and explore the variance explained by the spatial clustering of lambs on the same plots (measured as the conditional $R^2$ in comparison to the marginal $R^2$ coefficient of determination, computed with the *MuMIn* package). We used the posthoc Tukey test computed with the *multcomp* package for pairwise comparison of monthly estimates. We used equivalent GLMMs to explore possible differences in the log-transformed egg counts of *N. battus* and strongyles in relation to the time of sampling and sward type as linear predictors and plot ID as random grouping effect.

For GM composition (beta diversity) analysis, we filtered the 'raw' ASV data table to remove all ASVs recorded in < 4 individuals (corresponding to a prevalence of 5%) and with less than < 115 total reads (corresponding to < 1% of the minimum library size) to remove potential environmental transients from the data. We then computed the average Bray-Curtis dissimilarity index for each pair of samples (based on iteratively subsampled ASV tables according to smallest library size according to *avgdist()* function from vegan package). To explore whether overall GM composition varied in relation to sward type, time of sampling, FEC of *N. battus* and strongyles, as well as individual lamb identity, a marginal permutational analysis of variance (PERMANOVA) was performed on pairwise Bray-Curtis dissimilarity using the *adonis2()* function from the *vegan* package. Plot ID was included in the models as stratification effect to account for repeated sampling from the same plots. We quantified the distance-to-centroid dispersion of Bray-Curtis dissimilarity measures and tested for possible variation/heterogeneity of dispersion among different months of sampling and sward types using the *betadisper()* function in *vegan*. The similarity in GM composition from different samples was visualised by generating a Nonmetric multidimensional scaling (NMDS) plot based on Bray-Curtis dissimilarity as a measure of taxa turnover. Since preliminary analysis of Bray Curtis versus (taxonomically) weighted Unifrac metric did not reveal any different results, we focused on Bray-Curtis dissimilarity only.

For exploring changes in the relative abundance of the most common bacterial genera, in relation to sward type, time of sampling and FEC of *N. battus* and strongyles, we aggregated the ASV data to genus level (for those identified after the taxonomic cleaning, all others assigned to the group 'others') that were recorded in at least 30% of GM samples (corresponding to 18 samples) and with a relative abundance of ≥ 5% of the minimum library size (corresponding to ≥ 557 total reads). We then fitted a generalised linear latent variable model (GLLVM) using the *gllvm* package [38] to explore the variation in the relative abundance of common taxa in relation to predictor variables. GLLVMs comprise a multivariate model of joint species occurrence/abundances, whereby latent variables account for some of the variation in the data attributes to correlation in species abundances [38]. Microbiome data are compositional [39], and GLLVM offer a framework to analyse the compositions of microbiomes based on 'raw' (untransformed) sequencing table, since sampling bias arising

from unequal library sizes can be conveniently modelled as an 'offset' term as commonly practiced in count data regression models. We therefore implemented a GLLVM with a log-link negative binomial distribution, fixed row effects (corresponding to library size for each GM sample as offset term), and two latent variables (corresponding to two randomly fitted factors with taxa-specific factor loadings to capture compositional GM variance-covariance, [38]) and sward type, month of sampling, and *N. battus* and strongyle FECs as covariates. We fitted a GLLVM with the model structure described above (i.e., with predictor variable to data of ASV counts aggregated at family level for the most common families (n = 46 families recorded in at least 30% of samples with a relative abundance of ≥ 5% of the minimum library size; all others were amalgamated into an 'others' group). As for the GLLVM model output, we considered estimates to be 'significant' if the 95% confidence intervals (CI) did not overlap zero, and we reported differential abundance shifts as notable.

Moreover, we extracted the residual covariance matrix from the GLLVM fitted at family-level. This covariance matrix captures the correlation in co-occurrence and correlated abundance shifts [38], although we emphasise that due to the technical implementation of computing symmetric correlation coefficients among pairs of families, such co-occurrence coefficients do not necessarily allow insights into the true biological interactions effects but rather indicates co-occurrence patterns [40].

## Results

In total, we found 6,853 unique ASVs, of which 22 ASVs (0.3%) were found in all 60 faecal gut microbiome (GM) samples and 386 (6%) ASVs were found in at least 50 GM samples or more. At individual sample-level, we found on average 751 ± 24 (one standard error, SE) with a minimum of 277 unique ASVs in the different GM samples. These ASVs belonged to 222 identifiable genera and 116 identifiable families. The six most frequently detected identifiable genera (based on total reads) were: *Bacteroides* (5.9% of all reads), *Eubacterium* (5.1% of all reads), *Alistipes* (3.7% of all reads), *Ruminococcus* (2.8% of all reads), *Monoglobus* (1.9% of all reads), *Akkermansia* (1.2% of all reads), and *Treponema* (1.1% of all reads) (S1 Fig). The three most frequently detected identifiable families were: *Oscillospiraceae* (17.7% of all reads), *Lachnospiraceae* (10.2% of all reads), *Rikenellaceae* (9.3% of all reads). At the phylum level, the GM compositions were strongly dominated by bacteria of the phylum *Firmicutes*, which comprised on average 65% (min: 49%, max: 78%) of all reads (Fig 1). The second most dominant phyla were the *Bacteroidota*, which comprised on average 25% (min: 16%, max: 40%) of all reads. All other phyla were recorded with averaged relative abundances < 10%, while two different individuals

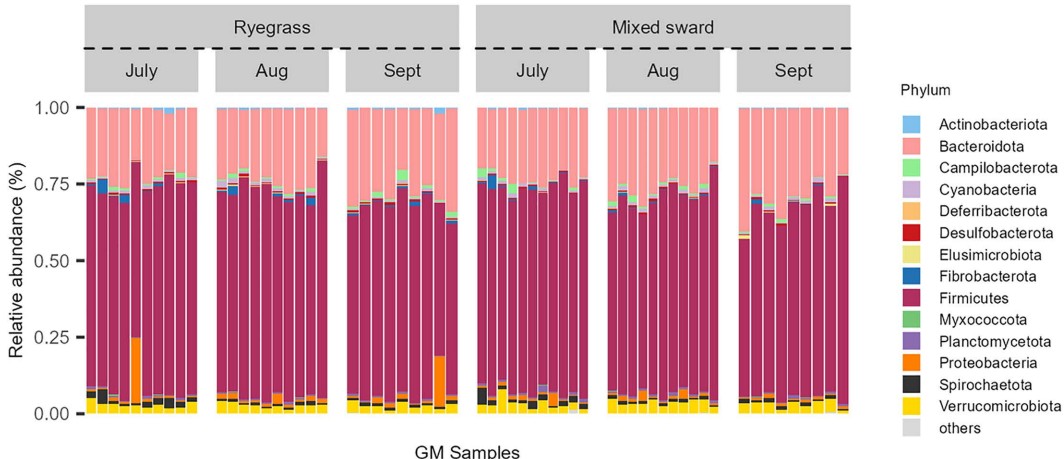

**Fig 1. Composition of gut microbiota in lambs grazing on two sward types across three consecutive months.** Stacked bar plots show the relative abundance of 16S rRNA gene sequences assigned to bacterial phyla in lambs grazing on ryegrass or mixed swards.

grazed on ryegrass showed elevated relative abundances of *Proteobacteria* with 21% and 16%, respectively (Fig 1, refer to S2 Fig for family-level plot).

## GM alpha diversity peaked in August amid large individual variation

The per-capita Chao1 species richness estimates ranged from 277 (CI: 281–275) to 1,097 (CI: 1139–1062) ASVs (**Fig 2**). The average per-capita Chao1 species richness was estimated to be 144 ± 44 ASVs (one SE) higher in August than in July (GLMM/Tukey $p < 0.01$) with substantial unexplained variation among samples (GLMM$_{Chao1}$ marginal $R^2 = 0.12$, conditional $R^2 = 0.45$). Likewise, the average per-capita Shannon diversity was estimated to be higher in August than in July (GLMM/Tukey $p < 0.01$), but the given predictors explained only a small fraction of the variation in GM Shannon diversity (GLMM$_{Shannon}$ marginal $R^2 = 0.13$, conditional $R^2 = 0.36$). We found no evidence of a relationship between alpha diversity measures and sward type, nor with faecal egg counts (FEC).

## GM composition varied (weakly) among sward types and time of sampling

Bray-Curtis dissimilarity as a measure of GM compositional dissimilarity ranged between 0.4 and 0.7 (mean 0.58 ± 0.001 one SE). GM composition differed with time of sampling and sward type (PERMANOVA 'time of sampling': F = 1.74, $p < 0.01$, 'sward type': F = 2.72, $p < 0.01$), but these factors explained only a small proportion of the variations in GM composition ('time of sampling': $R^2 = 0.055$, 'sward type': $R^2 = 0.043$). In contrast, individual lamb identity accounted for a larger proportion of variance (PERMANOVA, 'lamb ID': $R^2 = 0.40$). No evidence was found that the dispersion of GM compositions differed among sward types or time of sampling (dispersion tests $p = 0.47$ and $p = 0.16$, respectively). The NMDS ordination revealed some clustering of GM samples by treatment groups, including clustering among replicates from the same individuals (**Fig 3**).

## Gastrointestinal nematode burdens were higher in mixed swards

All lambs were infected with gastrointestinal nematodes (GIN): *Nematodirus battus* eggs were recorded with an overall prevalence of 73% (43 out of 59 samples with positive FEC) and strongyles were recorded with an overall prevalence of 98% (58 out of 59 samples with positive FEC). The average estimated eggs count of *N. battus* and strongyles were higher in lambs that grazed the mixed swards than those that grazed the ryegrass swards (both GLMM/Tukey comparisons $p < 0.05$). The average of estimated eggs burden of *N. battus* were higher in August than in July and September (both

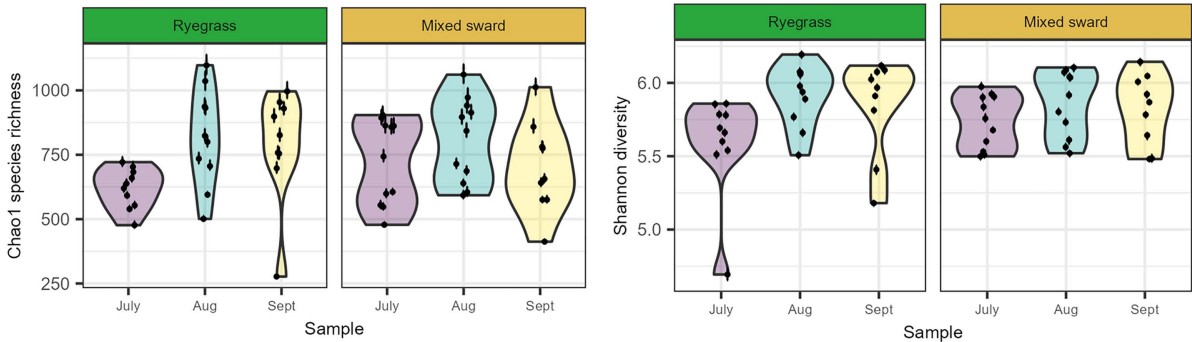

**Fig 2. Alpha diversity of gut bacterial assemblages in lambs grazing on different sward types across three months.** Per-capita Chao1 species richness (left panel) and Shannon diversity (right panel) estimates are shown for faecal gut bacterial assemblages from individual lambs sampled in different sward types and months. Points represent mean values, and bars indicate 95% confidence intervals, calculated from rarefied subsamples of the underlying amplicon sequence variant (ASV) tables. Grey triangles in the left panel denote the total number of ASVs observed in each host individual.

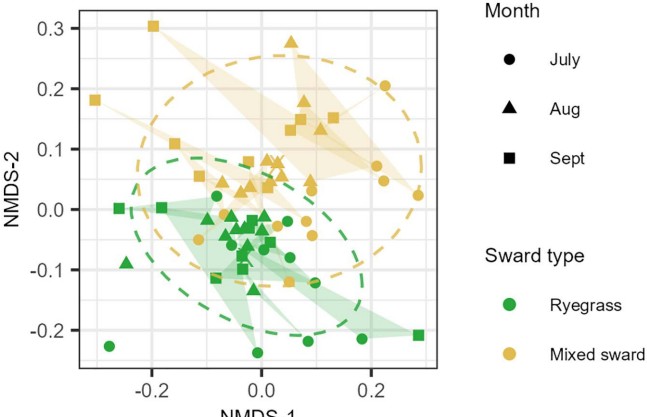

**Fig 3. Ordination of gut microbiome composition in lambs grazing on two sward types across three consecutive months.** Gut microbiome (GM) composition is visualized using nonmetric multidimensional scaling (NMDS) based on Bray–Curtis dissimilarities among samples collected from lambs grazing on ryegrass or mixed swards over three consecutive months. Each point represents a unique GM sample (n = 60 from 21 individuals). Ellipses indicate the 95% confidence intervals of group-level ordination spaces for each sward type, and shaded polygons connect samples originating from the same individual lamb.

pairwise GLMM/Tukey comparisons p < 0.05). Time of sampling and sward type explained about a third of the variation in FEC of *N. battus*, whereas the spatial clustering on plots explained no additional variation in FEC (GLMM$_{Nematodirus}$ marginal $R^2 = 0.11$, conditional $R^2 = 0.46$). Strongyle FEC were smallest in July after anthelminthic treatment (GLMM$_{Strongyles}$/Tukey p = 0.08).

## Microbial abundance shifts relate to sward type, time, and nematode burden

For the most common genera (n = 51), we found shifts in the relative abundance for 65 out of 255 (25%) genera-predictor relationships with regard to sward type, time of sampling, and faecal egg counts (**Fig 4**, refer to Supporting Information S1 Table for estimates).

**Sward type:** In response to sward type, we found one genus to be relatively less abundant (*Saccharofermentans*) and four genera (*Alistipes, Candidatus Soleaferrea, Monoglobus, Oscillibacter*) to be relatively more abundant in lambs grazing mixed swards compared to those grazing ryegrass (S1 Table). A prominent decrease in relative abundance was observed for the genus *Saccharofermentans* (*Oscillospiraceae*) in that the expected relative abundance of this genus was only 33% (CI: 19%−57%) of that in lambs grazing ryegrass.

**Sampling time:** We found the shifts in the relative abundance of common genera between July and August with 15 genera being relatively less abundant and six genera being relatively more abundant in July compared to August (S1 Table). Seventeen genera were relatively less abundant and three genera were relatively more abundant in September than in August (S1 Table). A prominent decrease in relative abundance was observed for the genus *Acetobacter* in that its relative abundance dropped in August to 22% (CI: 9%−56%) of that found in July. The abundances of *Negativibacillus* dropped in September to 9% (CI: 2%−37%) and of *Solibacillus* to 33% (CI: 22%−50%) to those found in August, respectively. The abundances of *Anaerosporobacter* increased in September 7 times (CI: 2.8–17.8) and of *Butyrivibrio* 6 times (CI: 2.5–14.5) to those found in August, respectively.

**Faecal egg count:** With larger FEC of *N. battus*, we found seven genera (*Bacteroides, Candidatus Saccharimonas, Candidatus_Soleaferrea, Colidextribacter, Desulfovibrio, Negativibacillus, Oscillibacter*) to be relatively less abundant and three genera (*Anaerosporobacter, Lysinibacillus, Romboutsia*) to be relatively more abundant (S1 Table). With larger FEC of strongyles, we found five genera (*Acetitomaculum, Anaerovorax, Helicobacter, Lysinibacillus, Olsenella*)

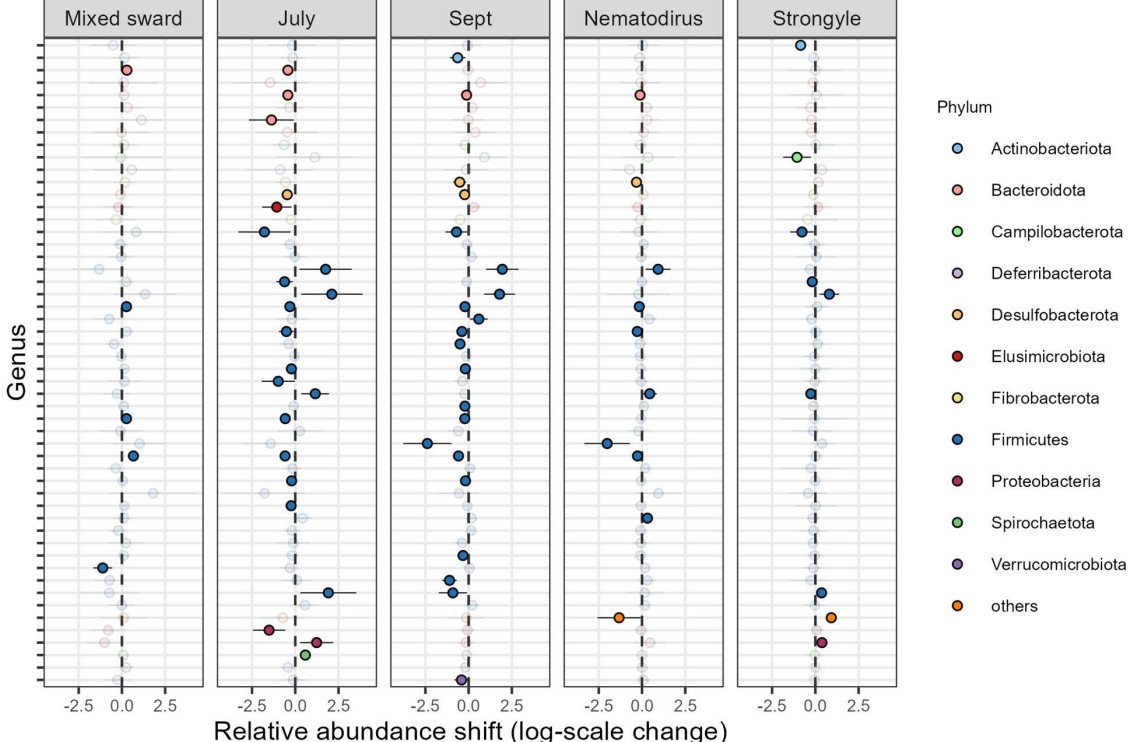

**Fig 4. Shifts in the relative abundance of common bacterial genera in lamb gut microbiomes across sward types and sampling months.** Shifts in the relative abundance of 47 common bacterial genera are shown, represented by the 616 most prevalent amplicon sequence variants (ASVs; occurring in ≥30% of samples and contributing ≥5% of the minimum library size). Coefficient estimates were derived from a generalized linear latent variable model (GLLVM) with a negative binomial distribution. Each darkly coloured point indicates a significant effect (95% confidence interval not overlapping zero), with point colours corresponding to bacterial phyla. The model baseline categories were ryegrass (for sward type) and August (for month).

to be relatively less abundant and four genera (*Butyrivibrio*, *Candidatus Saccharimonas*, *Escherichia-Shigella*, *Streptococcus*) to be relatively more abundant (S1 Table). A prominent decrease in relative abundance was observed for the genus *Negativibacillus* in that the expected relative abundance of this genus decreased by 13% (95% CI: 4%−50%) per one standard deviation increase in log-scale FEC of *N. battus*. The relative abundance of *Candidatus Saccharimonas* increased 2.5 times (CI: 2.1–3) per one standard deviation increase in log-scale FEC of strongyles and the relative abundance of *Olsenella* decreased by 43% (CI:34−0.54) per one standard deviation increase in log-scale FEC of strongyles.

Repeating the differential abundance analysis for the most common families (n = 45), we found shifts in the relative abundance for 55 out of 225 tested family-predictor relationships (Fig 5, Supporting Information S2 Table). More specifically, In July, we found nine families to be relatively less abundant and four families to be relatively more abundant than in August (S2 Table). In September, we found nine families to be relatively less abundant and nine families to be relatively more abundant than in August (S2 Table). In response to sward type, we found two families to be relatively less abundant and five families to be relatively more abundant in lambs grazing on mixed swards compared to those grazing on ryegrass (S2 Table). With larger faecal egg count of *N. battus*, we found five families to be relatively less abundant and five families to be relatively more abundant (S2 Table). With larger faecal egg count of strongyles, we found five families to be relatively less abundant and two families to be relatively more abundant (S2 Table).

                                                        

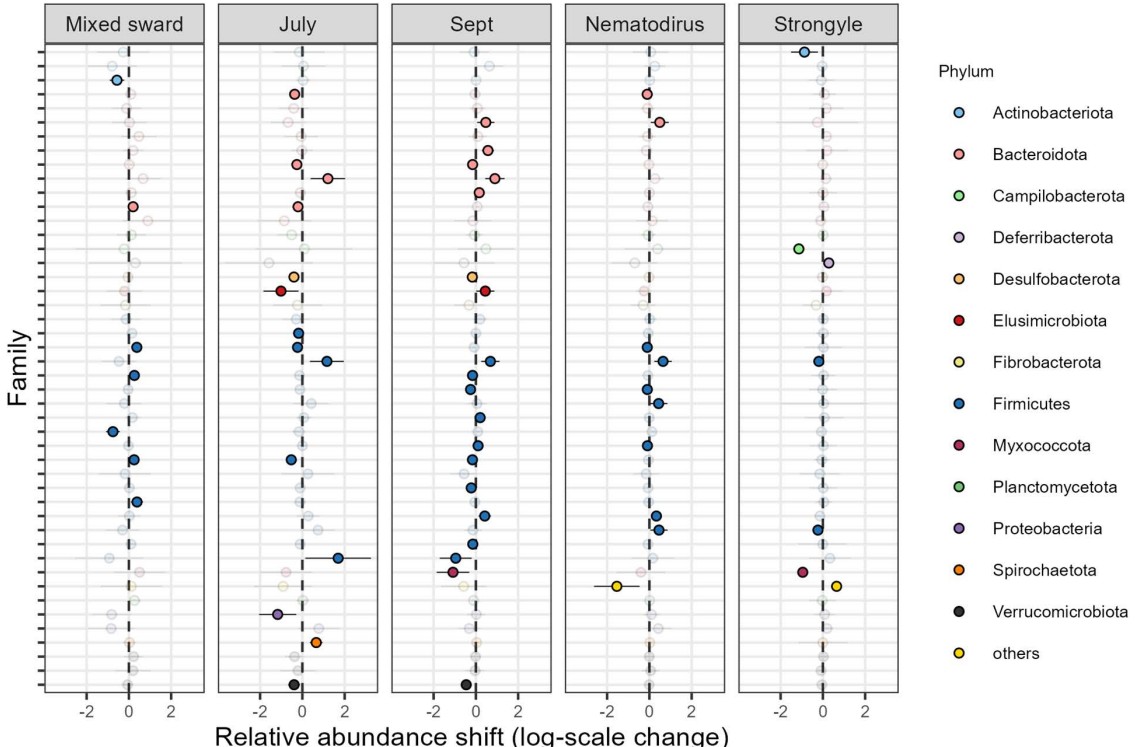

**Fig 5. Shifts in the relative abundance of common bacterial families in lamb gut microbiomes across sward types and sampling months.**
Shifts in the relative abundance of 52 common bacterial families (plus one grouped as "others") are shown. Families included occurred in ≥30% of samples and contributed ≥5% of the minimum library size. Coefficient estimates were obtained from a generalized linear latent variable model (GLLVM) with a negative binomial distribution. Each darkly coloured point indicates a significant effect (95% confidence interval not overlapping zero), whereas transparent points denote non-significant estimates. Point colours correspond to bacterial phyla. The model baseline categories were ryegrass (for sward type) and August (for month).

We found family-level correlations in the shifts in the relative abundance (according to correlation coefficients >0.7 from the residual covariance matrices of GLLVM) for four out of the 45 most common bacterial families. In particular, shifts in the relative abundance of *Bacillaceae* correlated with shifts of *Clostridiaceae*, *Peptostreptococcacea*, and *Planococcaceae*. Moreover, shifts in the relative abundance of *Clostridiaceae* correlated with abundance shifts of *Planococcaceae*.

## Discussion

Understanding the plasticity of animal microbiomes in response to environmental conditions and gastrointestinal parasitic infections is critical for grazing livestock, which depend on microbial symbionts for nutrient processing and overall health due to their extensive metabolic capabilities. In our study, we found clear signatures of sward type, time of sampling and gastrointestinal nematode (GIN) burden in structuring the faecal microbial assemblages in lambs, whereby individual host identity emerged as a prominent predictor of faecal gut microbiome (GM) compositional plasticity. Moreover, we identified distinct shifts in the relative abundance of microbial taxa associated with sward type, time of sampling, and faecal egg count (FEC) of *Nematodirus* and strongyles. At the same time, higher GIN burdens were observed in lambs grazing on mixed swards compared to those grazing on less diverse ryegrass pastures. These findings provide evidence that parasite-microbiome relationships are also shaped by underlying pasture conditions, pointing to potential triadic relationships between pasture condition, gastrointestinal parasites, and the GM.

## Gut microbiome composition is 'modestly' shaped by pasture and time

Grazing on a more diverse mixed sward, compared to a ryegrass-dominated pasture, resulted in clear but relatively 'modest' differences in the gut microbiome composition. These differences included changes in overall structural diversity and five differentially abundant bacterial genera, with the most notable change being a marked decline in the relative abundance of the genus *Saccharofermentans* (*Oscillospiraceae*). This outcome is at first glance somewhat surprising, given that a number of studies have suggested 'pronounced' effects of pasture and diet diversity on GM composition in sheep [41–44] and also in other ruminant species such as cattle, reindeer, and bison [8,22–24]. However, the relatively modest compositional changes we found are challenging to evaluate without understanding the functional implications, as even limited shifts in the abundance of functionally important taxa can have meaningful impacts. In our study, enriched taxa in mixed sward pastures included *Alistipes*, *Monoglobus*, and *Oscillibacter*. Among these taxa, *Alistipes* and *Oscillibacter* are broadly recognised in facilitating the anaerobic breakdown of complex molecules into short-chain fatty acids [41,45], which, in turn, are vital regulators of immunity, inflammation, and metabolism [46,47]. *Saccharofermentans*, in turn, which was the only genus more abundant in ryegrass-dominated pastures, is involved in saccharide metabolism and has been reported elsewhere to increase in preweaning calve rumen in response to gallic acid feeding [48]. While the functional consequences of the observed shifts in microbial abundance remain unknown, our findings are supporting the notion of previous studies demonstrating that sward type modulates gut microbial community structure by abundance shifts of taxa specialised in degrading distinct plant components; together these overall 'modest' alterations in microbial assemblages in response to sward type likely represent diet-driven adaptive plasticity, whereby the overall 'core' microbiome (here referring to microbial taxa commonly shared among lambs within the local population rather than a universally defined set) remains largely stable across lambs grazing different sward types [49].

In addition to pasture-driven effects, we observed temporal shifts in gut microbiome composition and the relative abundance of several bacterial genera. Some of these likely reflect adaptive responses to changes in diet composition or environmental factors affecting substrate availability. For instance, the anaerobic genus *Butyrivibrio* increased 6-fold from August to September. This genus is considered to play a role in fibre degradation and butyrate production [50], exemplifying how taxa involved in short-chain fatty acid metabolism may respond to environmental conditions. The pattern supports the notion that pasture conditions during the late grazing season, such as more mature, lignified plants, may favour hindgut microbes specialised in degrading complex carbohydrates. This is consistent with previous studies demonstrating that gut microbial communities in ruminants are highly dynamic and responsive to seasonal changes in forage quality [51,52]. At the same time, potential time lags in the response of microbial communities to dietary changes should be considered, as the establishment and stabilisation of bacterial populations may occur over extended time periods of several weeks rather than immediately following environmental shifts [53].

We anticipate that the simultaneous increases and decreases in the relative abundance of a number of taxa linked to seemingly similar functional roles (in our study, most abundance shift where found for genera involved in short-chain fatty acid metabolism) warrant caution when inferring functional implications from abundance shifts, as without further metabolomic profiling, the specific functional roles of individual species and potential functional redundancies remain unknown [54,55]. We also anticipate that abundance shift between July and August could be induced by the anthelminthic treatments that have previously found to shift microbial abundances in horses [56]. Notably, the 'time of sampling' variable encompassed several overlapping ecological and ecophysiological processes, including time since weaning, potential seasonal shifts in abiotic and biotic grazing conditions, and the post-treatment period following anthelmintic administration. Although such a combination of drivers would typically be expected to induce substantial shifts in microbiome composition, their cumulative effect on overall GM composition was nevertheless exceeded by the prominent influence of individual identity. If key microbial colonisation events occurred prior to the exposure to different sward types, it is plausible that the foundational communities were already well established in the lambs prior to weaning, such that subsequent dietary influence or any other aspects of pasture condition during the grazing season resulted mostly in differential abundance

shifts rather than bacterial species turnover. At the same time, the observed increase in species richness of gut bacteria between July and August suggests that at least some microbial taxa are acquired post-weaning during the first grazing season. Previous work on lamb microbiome diversity indicate that the early-life period, particularly the pre-weaning stage, is critical in establishing the 'core' microbiome [20,57]. Hence, if early life conditions are strong drivers of microbiome assemblage formation [58], it could be relevant for future work to explore whether well established microbiomes may be less responsive to environmental changes over the course of an individual's lifetime and how such stability links to the adaptive plasticity in response to dietary shift in relation to sward composition. Together, these findings highlight the interplay between host-specific factors and environmental inputs in shaping the GM and underscore the importance of considering individual-level variation and developmental timing in microbiome studies.

### Parasite-microbiome-environment relationships are complex

Since parasitic GINs share the same gut environment with intestinal microbiota, it is no surprise that parasite-microbiome reciprocal interactions are increasingly reported [15,45,59,60], especially as it is now widely understood that parasites can disrupt the stability of GM and lead to idiosyncratic configurations consistent with the Anna Karenina principle in that each dysbiotic microbiome is disrupted in its own unique way [61]. In our study, we found that GIN burdens differ in lambs grazing different sward types, while also being correlated to shifts in the relative abundance of gut bacteria, including the genera *Negativibacillus, Candidatus Saccharimonas,* and *Olsenella.*

The genus *Negativibacillus* has been suggested to be potentially pathogenic in livestock [62,63]. The observed strong negative correlation between *Negativibacillus* abundance and *N. battus* egg loads might indicate that lower abundance of this genus is associated with increased host resilience to gastrointestinal nematode (GIN) infection. Additionally, *Negativibacillus* was found to be less abundant in September, coinciding with significantly lower *N. battus* egg counts compared to August, but depicting an additional temporal component. Whether these patterns are indeed linked to potential pathogenic effect of *Negativibacillus* and immune response dynamics requires future research. The genus *Candidatus Saccharimonas*, which strongly increased in abundance with increasing strongyle egg count in our study, has been recently also reported as being a member of the gut microbiome of the strongyle *Haemonchus contortus* in sheep [64], highlighting just another dimension of the possible diversity of gastrointestinal parasite-microbiome relationships.

Moreover, out of the 17 different genera with abundance shift linked to GIN burden (either *N. battus* or strongyles egg counts), 13 of the genera shifted also in abundance in response to pasture condition or time of sampling, lending further support to the idea of possible triadic relationships between spatiotemporal variation in pasture condition, microbiomes and GIN burden.

Notably, the seemingly equal number of microbial genera affected negatively versus positively by GIN burden (seven versus three with regard to *N. battus* FEC and five versus four with regard to strongyle FEC) together with the likely involvement of a number of these genera in carbohydrate metabolism and the overall modest effect of GIN burden on assemblage composition, means that in the studied lamb population, there was no evidence of strong dysbiosis induced by increasing GIN burden. Rather, the simultaneous increases and decreases in the relative abundance of different genera warrant caution when inferring metabolic functions from abundance shifts, as without further metabolomic profiling, the specific functional roles of individual species and potential functional redundancies [54,55] remain unclear.

Our findings align with other studies showing that higher GIN burdens, rather than infection status alone, lead to shifts in the relative abundance of gut bacteria [14,29]. Unlike this previous work, we found some evidence that such parasite-microbiome relationships are also shaped by underlying pasture conditions, suggesting possible triadic relationship between gastrointestinal parasites, sward type/ diet, and the GM [65].

While the underlying mechanisms remain unexplored due to the correlative nature of our study, and the pragmatically motivated anthelmintic treatment at the study's onset introduced bias that limits inference about temporal changes in GIN burden, the higher GIN burden observed in lambs grazing mixed swards remains notable and aligns with patterns

reported in recent research. Mixed swards with taller and more heterogeneous vegetations structure can, for example, provide more stable microclimates that maintain moisture of faecal pats and the surrounding vegetation conditions that is crucial for parasite eggs and larvae development and survival in the off-host environment [66]. At the same time the higher plant diversity in mixed swards compared to ryegrass-based pastures may comprise more varied phytochemical profiles, which in turn could affect microbial assemblages and helminths alike through shift in nutrition availability or antimicrobial and anthelmintic properties [67–69], whereby a potential increase in anthelminthic properties is contradicting the higher observed pathogen load in mixed swards in our study. Other possible drivers of triadic parasite-microbiome-diet relationships could be changes to the within-host environment, including gut physiology and immune responses [25,70] or difference in microclimate that could impact the hatching of larvae from parasite eggs [71], which remain unknown for our study system.

Taken together, our study demonstrates that individual identity can be a stronger predictor of variation in GM composition than sward type in grazing lambs but even if the GM composition shows limited relationships with environmental conditions, environmental exposure in synergy with GINs can be linked to substantial shift in the relative abundance of specific bacterial taxa. Our findings emphasise the need for integrative studies that consider both ecological and pathological contexts when assessing microbiome dynamics and the possible functional implications in livestock. For healthier and more sustainable animal management practices, understanding the triadic interactions between livestock grazing on different swards, parasitic infections and microbial communities will be crucial for designing effective pasture management systems that simultaneously mitigate parasitic burdens and promote functionally beneficial microbiomes.

## Supporting information

**S1 Fig. Rank–abundance distribution of bacterial genera across all lamb gut microbiome samples.** Rank–abundance distribution of bacterial genera based on the total number of reads from all 60 samples. Genera are ranked by their total read counts, and their relative abundances are displayed on a logarithmic scale. Labels identify the six most abundant genera and the aggregated "other" group, which includes all ASVs not assigned to a genus level. (TIFF)

**S2 Fig. Gut microbiota composition at the family level in lambs grazing on different sward types across three consecutive months.** Gut microbial (GM) composition in lambs grazing on two sward types (ryegrass and mixed sward) over three consecutive months. Stacked bar plots show the relative abundance of 16S rRNA gene sequences assigned to different bacterial families. A total of 116 identifiable families were detected, including *Oscillospiraceae, Bacteroidaceae, Rikenellaceae, Campylobacteraceae, Acidaminococcaceae, Enterobacteriaceae, Prevotellaceae, Lachnospiraceae, Akkermansiaceae, Ruminococcaceae, Christensenellaceae, Butyricicoccaceae, Pirellulaceae, Planococcaceae, Monoglobaceae, Desulfovibrionaceae, Anaerovoracaceae, Spirochaetaceae, Moraxellaceae, Weekselaceae, Hungateiclostridiaceae, Peptostreptococcaceae, Fibrobacteraceae, Defluviitaleaceae, Erysipelotrichaceae, Peptococcaceae, Helicobacteraceae, Bacillaceae, Victivallaceae, Paludibacteraceae, Bifidobacteriaceae, Barnesiellaceae, Clostridiaceae, Marinifilaceae, Flavobacteriaceae, Muribaculaceae, Eggerthellaceae, Myxococcaceae, Atopobiaceae, Elusimicrobiaceae, Marinilabiliaceae, Deferribacteraceae, Streptococcaceae, Mycoplasmataceae, Sphingomonadaceae, Acholeplasmataceae, Acetobacteraceae, Tannerellaceae, Sphingobacteriaceae, Oxalobacteraceae, Saccharimonadaceae, Micrococcaceae, Corynebacteriaceae, Staphylococcaceae, Caulobacteraceae, Oligosphaeraceae, Aerococcaceae, Puniceicoccaceae, Enterococcaceae, Carnobacteriaceae, Nocardioidaceae, Hymenobacteraceae, Comamonadaceae, Endomicrobiaceae, Anaerolineaceae, Rhodobacteraceae, Dysgonomonadaceae, Microbacteriaceae, Rhizobiaceae, Pseudomonadaceae, Xanthomonadaceae, Selenomonadaceae, Beijerinckiaceae, Caldicoprobacteraceae, Geodermatophilaceae, Eubacteriaceae, Dermabacteraceae, Paenibacillaceae, Sutterellaceae, Dietziaceae, Succinivibrionaceae, Nitrosomonadaceae, Porphyromonadaceae, Hydrogenophilaceae, Veillonellaceae, Chitinophagaceae,*

*Intrasporangiaceae, Terasakiellaceae, Xanthobacteraceae, Devosiaceae, Nocardiaceae, Cellvibrionaceae, Lactobacilla-ceae, Chlamydiaceae, Synergistaceae, Coriobacteriaceae, Actinomycetaceae, Paracaedibacteraceae, Dermacoccaceae, Azospirillaceae, Coxiellaceae, Deinococcaceae, Longimicrobiaceae, Nakamurellaceae, Fusobacteriaceae, Microscilla-ceae, Blastocatellaceae, Bacteriovoracaceae, Syntrophomonadaceae, Isosphaeraceae, Solirubrobacteraceae, Chthonio-bacteraceae, Neisseriaceae, Thermaceae,* and *Diplorickettsiaceae.* Amplicon sequence variants (ASVs) not identifiable at the family level were grouped as "others.".
(TIFF)

**S1 Table. Significant predictor effects on the differential abundance of common bacterial genera in lamb gut microbiomes.** The table lists significant effects of predictors on the relative abundance of the most common bacterial genera, as estimated from a generalized linear latent variable model (GLLVM) with a negative binomial distribution. 'Significant' effects are defined as those whose 95% confidence intervals of the coefficient estimates do not overlap zero. Columns indicate the predictor (treatment: month, sward type, or parasite load), genus, family, phylum, and the model estimate (with 95% confidence interval in parentheses). Positive estimates indicate an increase in relative abundance relative to the model baseline, and negative estimates indicate a decrease.
(DOCX)

**S2 Table. Significant predictor effects on the differential abundance of common bacterial families in lamb gut microbiomes.** The table lists significant effects of predictors on the relative abundance of the most common bacterial families, estimated using a generalized linear latent variable model (GLLVM) with a negative binomial distribution. 'Significant' effects are defined as those whose 95% confidence intervals of the coefficient estimates do not overlap zero. Columns indicate the predictor (treatment: month, sward type, or parasite load), family, phylum, and the model estimate (with 95% confidence interval in parentheses). Positive estimates indicate an increase in relative abundance relative to the model baseline, while negative estimates indicate a decrease.
(DOCX)

## Acknowledgments

Parts of the computation was run on the Supercomputing Wales facility (supported by the European Regional Development Fund, ERDF, via Welsh Government).

## Author contributions

**Conceptualization:** Konstans Wells, Claire Reigate, M. Jordana Rivero, Tamsyn Uren Webster.

**Data curation:** Konstans Wells, Claire Reigate, M. Jordana Rivero, Tamsyn Uren Webster.

**Formal analysis:** Konstans Wells, Jack Jefferson, Tamsyn Uren Webster.

**Funding acquisition:** Konstans Wells, Claire Reigate, M. Jordana Rivero, Tamsyn Uren Webster.

**Methodology:** Konstans Wells, Jack Jefferson, Claire Reigate, Alessandra Giacomini, M. Jordana Rivero, Matthew Hitchings, Tamsyn Uren Webster.

**Project administration:** Konstans Wells, Tamsyn Uren Webster.

**Supervision:** Konstans Wells, Tamsyn Uren Webster.

**Writing – original draft:** Konstans Wells, Jack Jefferson, Tamsyn Uren Webster.

**Writing – review & editing:** Konstans Wells, Jack Jefferson, Claire Reigate, Alessandra Giacomini, M. Jordana Rivero, Matthew Hitchings, Tamsyn Uren Webster.

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
