## [Decision Letter · Decision Letter 0]

31 Oct 2025

Dear Dr. Wells,

Thank you for submitting your manuscript to PLOS ONE. After careful consideration, we feel that it has merit but does not fully meet PLOS ONE’s publication criteria as it currently stands. Therefore, we invite you to submit a revised version of the manuscript that addresses the points raised during the review process. In particular I recommend that you clarify in your manuscript that conclusions about individual differences among samples are based on the assumption that each sample comes from a different individual. However, this assumption does not apply to your experimental design.

Additionally, although microbial stabilization is reported to occur over several months (see Clemmons et al., 2019. Temporal Stability of the Ruminal Bacterial Communities in Beef Steers. Scientific Reports, 9, 9522), it may be useful to mention this reference when discussing the timeline of microbial shifts, even if your design did not allow for true longitudinal individual tracking.

We look forward to receiving your revised manuscript.

Kind regards,

Emmanuel Serrano, PhD

Academic Editor

PLOS ONE

“We thank the Ecological Continuity Trust for a grant that supported the laboratory work and the BSAS Steve Bishop Net Zero Award for supporting field research and lab analysis. We thank the Rothamsted Farm Staff for support during field work with support from Techion UK Ltd. The grazing experiment is part of the Institute Strategic Programme "AgZero+: Towards sustainable, climate-neutral farming" (NE/W005050/1), an initiative jointly supported by NERC and BBSRC. Parts of the computation was run on the Supercomputing Wales facility, part-funded by the European Regional Development Fund (ERDF) via Welsh Government.”

“We thank the Ecological Continuity Trust for a grant that supported the laboratory work and the BSAS Steve Bishop Net Zero Award for supporting field research and lab analysis. We thank the Rothamsted Farm Staff for support during field work with support from Techion UK Ltd. The grazing experiment is part of the Institute Strategic Programme "AgZero+: Towards sustainable, climate-neutral farming" (NE/W005050/1), an initiative jointly supported by NERC and BBSRC. Parts of the computation was run on the Supercomputing Wales facility, part-funded by the European Regional Development Fund (ERDF) via Welsh Government.”

“We thank the Ecological Continuity Trust for a grant that supported the laboratory work and the BSAS Steve Bishop Net Zero Award for supporting field research and lab analysis. We thank the Rothamsted Farm Staff for support during field work with support from Techion UK Ltd. The grazing experiment is part of the Institute Strategic Programme "AgZero+: Towards sustainable, climate-neutral farming" (NE/W005050/1), an initiative jointly supported by NERC and BBSRC. Parts of the computation was run on the Supercomputing Wales facility, part-funded by the European Regional Development Fund (ERDF) via Welsh Government.”

Additional Editor Comments:

Dear Dr. Konstants and colleagues,

The referee and I found your work timely and well-written, despite the limitations associated with the lack of individual identification of samples and the administration of anthelmintic treatments.

I recommend that you clarify in your manuscript that conclusions about individual differences among samples are based on the assumption that each sample comes from a different individual. However, this assumption does not apply to your experimental design.

Additionally, although microbial stabilization is reported to occur over several months (see Clemmons et al., 2019. Temporal Stability of the Ruminal Bacterial Communities in Beef Steers. Scientific Reports, 9, 9522), it may be useful to mention this reference when discussing the timeline of microbial shifts, even if your design did not allow for true longitudinal individual tracking.

Best regards,

Emmanuel SERRANO

Reviewers' comments:

Reviewer's Responses to Questions

**Comments to the Author**

1. Is the manuscript technically sound, and do the data support the conclusions?

Reviewer #1: Yes

2. Has the statistical analysis been performed appropriately and rigorously?

Reviewer #1: Yes

3. Have the authors made all data underlying the findings in their manuscript fully available?

Reviewer #1: Yes

4. Is the manuscript presented in an intelligible fashion and written in standard English?

Reviewer #1: Yes

Reviewer #1: In this study, the authors examine associations between lamb gut microbiome diveristy and composition, and gastintestinal nematodes, pasture type and time of year, utilising 16S metabarcoding. They demonstrate pasture type has a modest impact on bacterial community diversity and composition, with shifts in abundance of specific genera. In addition, they find key differences in gut microbial communties between lambs with differeing nematode worm burdens, and across monthly sampling periods. Overall, this provide novel information on the potential interactive effects of diet, environmental and parasites on lamb gastrointestinal microbial communities. The manuscript is largely clearly written, the methods robust and statistical approaches appropriate for the data. I only have a couple of very minor comments:

Methods: Please provide information on the use of negative controls for the molecular work. Did you have DNA extraction controls and PCR negative controls? How were control data used bioinformatically?

L150: "mostly not be identified to species level", should it read most could not be?

L389-394: Long/hard to read sentence, please break up for clarity.

**Do you want your identity to be public for this peer review?** For information about this choice, including consent withdrawal, please see our Privacy Policy

Reviewer #1: No

---

## [Author Response · Author response to Decision Letter 1]

3 Nov 2025

Point-Point Responses PONE-D-25-39654

(line number refer to those in the track-change file version)

Editor (Dr Emmanual Serrano)

The referee and I found your work timely and well-written, despite the limitations associated with the lack of individual identification of samples and the administration of anthelmintic treatments.

I recommend that you clarify in your manuscript that conclusions about individual differences among samples are based on the assumption that each sample comes from a different individual. However, this assumption does not apply to your experimental design.

Response: We thank the editor for pointing this important detail out. In anticipation of the editor’s feedback, we have further clarified that all faecal samples were collected from individually marked lambs. This approach ensured that samples obtained within the same sampling period represented different individuals, whereas the lambs were repeatedly sampled across the three-month study period. We have now revised the description of the sample collection as “We analysed 60 freshly voided faecal samples collected non-invasively from 21 weaned, individually marked lambs grazing on four experimental plots. Samples were collected on three dates between July and September 2023 (20th July, 20th August, 13th September), with 21 samples each in July and August, and 18 in September. While the same marked lambs were repeatedly sampled over time, the final dataset comprised samples randomly selected from a larger pool, and therefore did not include exactly the same individuals across all sampling occasions.” (lines 115-120).

Additionally, although microbial stabilization is reported to occur over several months (see Clemmons et al., 2019. Temporal Stability of the Ruminal Bacterial Communities in Beef Steers. Scientific Reports, 9, 9522), it may be useful to mention this reference when discussing the timeline of microbial shifts, even if your design did not allow for true longitudinal individual tracking.

Response: Thank you. We have now considered this reference and highlighted the delay in microbiome shifts and stabilisation in response to changing diet etc as “At the same time, potential time lags in the response of microbial communities to dietary changes should be considered, as the establishment and stabilisation of bacterial populations may occur over extended time periods of several weeks rather than immediately following environmental shifts [53]” (lines 402-405).

Referee: 1

In this study, the authors examine associations between lamb gut microbiome diveristy and composition, and gastrointestinal nematodes, pasture type and time of year, utilising 16S metabarcoding. They demonstrate pasture type has a modest impact on bacterial community diversity and composition, with shifts in abundance of specific genera. In addition, they find key differences in gut microbial communities between lambs with differing nematode worm burdens, and across monthly sampling periods. Overall, this provide novel information on the potential interactive effects of diet, environmental and parasites on lamb gastrointestinal microbial communities. The manuscript is largely clearly written, the methods robust and statistical approaches appropriate for the data. I only have a couple of very minor comments:

Methods: Please provide information on the use of negative controls for the molecular work. Did you have DNA extraction controls and PCR negative controls? How were control data used bioinformatically?

Response: We much appreciate the positive and thoughtful feedback of the reviewer – thank you!

We have now stated how we used negative controls in the manuscript, reading as “Two negative controls (PCR library blanks) were prepared and sequenced alongside the faecal DNA samples. The negative controls each yielded < 300 reads and were therefore excluded from subsequent analysis” (lines 179-181).

L150: "mostly not be identified to species level", should it read most could not be?

Response: We rewrote the respective sentence as “All other nematode eggs pooled into a single strongyle-type FEC category (Strongyloides and related genera), because most could not be reliably identified to the species level” (lines 154-155). This should now clarify that most eggs from other taxa could not be identified to species level.

L389-394: Long/hard to read sentence, please break up for clarity.

Response: We have shorted the sentences and revised the section as “In addition to pasture-driven effects, we observed temporal shifts in gut microbiome composition and the relative abundance of several bacterial genera. Some of these likely reflect adaptive responses to changes in diet composition or environmental factors affecting substrate availability. For instance, the anaerobic genus Butyrivibrio increased 6-fold from August to September. This genus is considered to play a role in fibre degradation and butyrate production [50], exemplifying how taxa involved in short-chain fatty acid metabolism may respond to environmental conditions. The pattern supports the notion that pasture conditions during the late grazing season, such as more mature, lignified plants, may favour hindgut microbes specialized in degrading complex carbohydrates.” (lines 386-393).

---

## [Editor Report · Decision Letter 1]

4 Nov 2025

Triadic relationships between pasture exposure, gastrointestinal parasites, and hindgut microbiomes in grazing lambs

PONE-D-25-39654R1

Dear Dr. Wells,

We’re pleased to inform you that your manuscript has been judged scientifically suitable for publication and will be formally accepted for publication once it meets all outstanding technical requirements.

Kind regards,

Emmanuel Serrano, PhD

Academic Editor

PLOS ONE

Additional Editor Comments (optional):

My congratulations

Emmanuel
---

## [Editor Report · Acceptance letter]

PONE-D-25-39654R1

PLOS ONE

Dear Dr. Wells,

I'm pleased to inform you that your manuscript has been deemed suitable for publication in PLOS ONE. Congratulations! Your manuscript is now being handed over to our production team.

Kind regards,

on behalf of

Dr. Emmanuel Serrano

Academic Editor

PLOS ONE